# Multiscale compression-induced restructuring of stacked lipid bilayers: From buckling delamination to molecular packing

**Marilyn Porras-Gómez**[1],[◎], **Hyunchul Kim**[2],[◎], **Mohan Teja Dronadula**[3], **Nurila Kambar**[1], **Christopher J. B. Metellus**[2], **Narayana R. Aluru**[3], **Arend van der Zande**[2]*, **Cecília Leal**[1]*

**1** Department of Materials Science and Engineering, University of Illinois Urbana-Champaign, Urbana, Illinois, United States of America, **2** Department of Mechanical Science and Engineering, University of Illinois Urbana-Champaign, Urbana, Illinois, United States of America, **3** Walker Department of Mechanical Engineering, Oden Institute for Computational Engineering and Sciences, The University of Texas at Austin, Austin, Texas, United States of America

◎ These authors contributed equally to this work.
* arendv@illinois.edu (AZ); cecilial@illinois.edu (CL)

**Data Availability Statement:** All relevant data are within the paper and its Supporting information files.

## Abstract

Lipid membranes in nature adapt and reconfigure to changes in composition, temperature, humidity, and mechanics. For instance, the oscillating mechanical forces on lung cells and alveoli influence membrane synthesis and structure during breathing. However, despite advances in the understanding of lipid membrane phase behavior and mechanics of tissue, there is a critical knowledge gap regarding the response of lipid membranes to micro-mechanical forces. Most studies of lipid membrane mechanics use supported lipid bilayer systems missing the structural complexity of pulmonary lipids in alveolar membranes comprising multi-bilayer interconnected stacks. Here, we elucidate the collective response of the major component of pulmonary lipids to strain in the form of multi-bilayer stacks supported on flexible elastomer substrates. We utilize X-ray diffraction, scanning probe microscopy, confocal microscopy, and molecular dynamics simulation to show that lipid multilayered films both in *gel* and *fluid* states evolve structurally and mechanically in response to compression at multiple length scales. Specifically, compression leads to increased disorder of lipid alkyl chains comparable to the effect of cholesterol on *gel* phases as a direct result of the formation of nanoscale undulations in the lipid multilayers, also inducing buckling delamination and enhancing multi-bilayer alignment. We propose this cooperative short- and long-range reconfiguration of lipid multilayered films under compression constitutes a mechanism to accommodate stress and substrate topography. Our work raises fundamental insights regarding the adaptability of complex lipid membranes to mechanical stimuli. This is critical to several technologies requiring mechanically reconfigurable surfaces such as the development of electronic devices interfacing biological materials.

**Funding:** AZ, NA NSF DMR-1720633 National Science Foundation through the University of Illinois Urbana-Champaign Materials Research Science and Engineering Center https://mrsec.illinois.edu AZ, NY NSF 2140225 National Science Foundation https://www.nsf.gov/ CL ONR N00014-21-1-2029 Office of Naval Research https://www.nre.navy.mil/ The funders had no role in study design, data collection and analysis, decision to publish, or preparation of the manuscript.

**Competing interests:** The authors have declared that no competing interests exist.

# Introduction

Natural membranes dynamically adjust their composition, shape, and form as a function of biochemical, biophysical, and mechanical cues. While the influence of mechanical properties on cells and tissues is a focus of numerous studies, the direct impact of micro-mechanical properties on membranes is far less explored [1]. However, membranes are subjected to significant strain in numerous biological processes. One relevant example is alveolar distension during breathing [2–4]. Alveolar basement membranes and epithelial cells can experience a mechanical stretch load from 4 to 12% linear distension in normal to deep breathing, and 25% in total lung capacity [2, 5]. In addition, distension can be rather heterogeneous even within a single alveolus, especially in diseased lungs.

It is widely recognized that mechanical stress influence the biological function and signaling of alveolar epithelial cells both in healthy and diseased states [6]. Specifically, mechanical stretch of cultured epithelial cells have resulted in increased permeability [7, 8] and enhanced pulmonary surfactant production [9–11]. More recently, such mechanical transduction pathways have been validated in reconstituted alveolar-capillary interfaces [12] where human epithelial and endothelial cell sheets where deposited onto polydimethylsiloxane (PDMS) substrates that were mechanically challenged up to 15% of uni-axial strain. However, the isolated effect that oscillations in mechanical properties have on pulmonary lipid membranes is still unclear. This is important because pulmonary lipid (often referred to as surfactant) membranes are the first barrier to oxygen intake at the air-liquid interface. Lung surfactant membranes are also the main material controlling alveolar surface tension, which supports most of the transpulmonary pressure [13–15].

Lipid bilayers are simple yet effective systems to study membrane mechanics [16, 17], structure [18] and phase behavior [19]. For example, the addition of cholesterol leads to the formation of phase-separated liquid-order $L_o$ and liquid-disordered $L_\alpha$ domains [20] The same result can be obtained upon the application of pressure, which tunes bilayer compressibility and thickness [21]. Moreover, lipid bilayers in the *gel*–phase exhibit exotic elastic behavior upon bending such as softening and discontinuous buckling leading to negative compressibility [22]. Supported lipid bilayers (SLBs) and even cell membranes supported onto flexible elastomer substrates grow lipid reservoirs in the form of vesicles and tubes – which minimizes elastic and adhesion energies, as a response to expansion and compression cycles [23–25]. Surface hydrophilicity [26] and topography [27] influence the mechanisms of lipid bilayer remodeling. Periodicities decrease the membrane adhesion energy –i.e. less favorable to adhere compared to a planar surface [27]. Lipid bilayers supported onto wrinkled elastomer substrates preferentially adhere to the largest wrinkles, and wrinkling induces changes in chain conformational order [28]. Wrinkled substrates also introduce one and two dimensional curvature onto planar lipid bilayers and cause domain reorganization of lipid mixtures [29]. Evidently, lipid bilayers supported onto flexible substrates exhibit mechanical behavior and adaptation mechanisms significantly different from those present in freestanding lipid bilayers [30], or even those present in lipid bilayers supported onto rigid substrates [31].

Most of the knowledge that we have on the response of lipid bilayers to mechanical cues were acquired using single bilayer systems. Yet, lipid multilayered systems are common, one important example being the multilamellar bodies (MLBs) in pulmonary membranes that are synthesized by alveolar type II cells. This process is strongly regulated by mechanical cues [9–11] and MLBs mechanically adapt to expansion and contraction during the breathing cycle [6, 32]. We know that phase-separation in single-bilayers is correlated across layers [33, 34], and that the effect of membrane additives in Young modulus of SLBs propagate to multilayers [35].

How multiple lipid layers collectively respond to stress and deformation, and how the effect is correlated across layers, remains unknown.

Here, we evaluate the emerging structural and mechanical behavior of the major lipid of pulmonary multilamellar bodies in response to compressive stress. Specifically, we study structure, morphology and adhesion of multilayered lipid systems in *gel* $L_\beta$ and *fluid* $L_\alpha$ phases, and how they evolve under compressive stress. Utilizing both experimental and simulation approaches, we observe four types of multiscale reconfiguration upon compression of the lipid multilayered membranes: 1) multilayer alignment while the order of hydrocarbon lipid chains decreases, 2) correlation of nanoscale defects across layers, 3) development of nanometer-scale film corrugations as well as microscale wrinkling and buckling delamination. Altogether, we describe the reconfiguration of lipid multilayered films across multiple lengthscales. Their dynamic behavior exhibited during compression has the potential to impact the development of emerging applications such as substrate-mediated bioelectronics.

## Results and discussion

We evaluate the multiscale structure and mechanics of supported phosphatidylcholine (PC) multilayered lipid films (SLM) under uniaxial compression. Fig 1A and 1B schematically shows the structure at different length scales of lipid films supported onto a polydimethylsiloxane (PDMS) substrate before and after applying compression. As we will demonstrate, the act of compressing the film induces both 3D buckle-delamination and wrinkling at the nano to microscales, as well as restructuring of the lipid layers at the molecular scale. In order to induce and control compression, we first pre-strained the PDMS substrates to 5–20% of the original length (S1 File). PDMS is a biocompatible elastomer with an elastic limit over 100%, and well-studied mechanics [36–38]. We spin coat deposit either dipalmitoylphosphatidylcholine (DPPC) or dioleoylphosphatidylcholine (DOPC) SLM films with thickness of 50–100 nm on the pre-strained PDMS. DPPC is the major component of pulmonary membranes and we choose these two PC lipids because they allow comparison of different phases under compression. Under ambient conditions, DPPC is stable in the $L_\beta$*gel* phase, while DOPC is stable in the $L_\alpha$*fluid* phase. During deposition and for all measurements, the SLM films are at room temperature and room relative humidity (RH: 40–60%). Finally, we released the substrate pre-strain

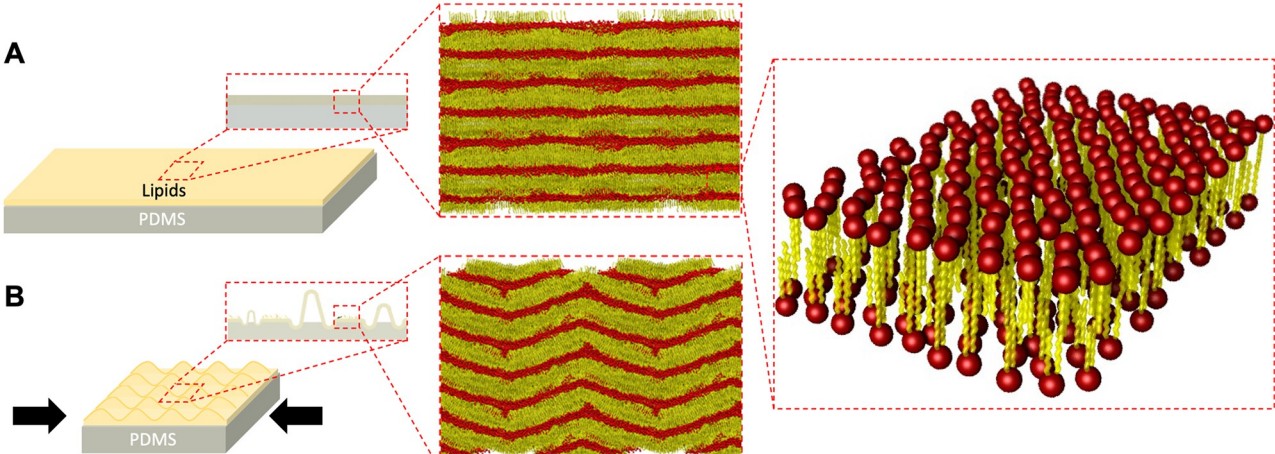

**Fig 1. Substrate-mediated reconfiguration of lipid multilayered films.** Schematics and simulations (discussed further in Fig 5) of the multiscale structure of multilayered lipid films on a deformable substrate before (A) and after (B) compression. The schematic in the right expands on a section of the lipid multilayered system to highlight a bilayer where headgroups are represented in red and tails in yellow.

by controlled increments of 5, 10 or 20%, to induce compression on the lipid film. To gain a complete picture of the multiscale structure and mechanics we combined grazing incidence small (GISAXS) and wide (GIWAXS) angle X-ray scattering, atomic force microscopy (AFM) topography, confocal laser scanning microscopy (CLSM) and coarse-grained molecular dynamics simulations.

## Compression induced reconfiguration of SLM structure

Fig 2 shows the effect of compression on DPPC SLM films in *gel* $L_\beta$ phase at room temperature ($T < T_m$) using grazing incidence small and wide angle X-ray scattering (GISAXS and GIWAXS respectively), which use a low angle of X-ray beam incidence relative to the substrate in order to enhance diffraction from thin layers. GIWAXS probes correlations at shorter range (molecular scale), while, GISAXS probes correlations at longer range (e.g. between multiple lipid bilayers). Fig 2A shows the GIWAXS diffraction patterns obtained for DPPC SLM films uncompressed (flat) and after compression of 10%, while Fig 2B plots the integrated intensity vs scattering vector $q$. Fig A in S2 File compares the corresponding GISAXS diffraction patterns, and Figure B shows the GIWAXS for just the PDMS substrate.

The position, width and shape of GISAXS and GIWAXS peaks provide information on the lattice parameters and crystalline order of the SLM. Fig 2B, shows a Lorentzian function fit to

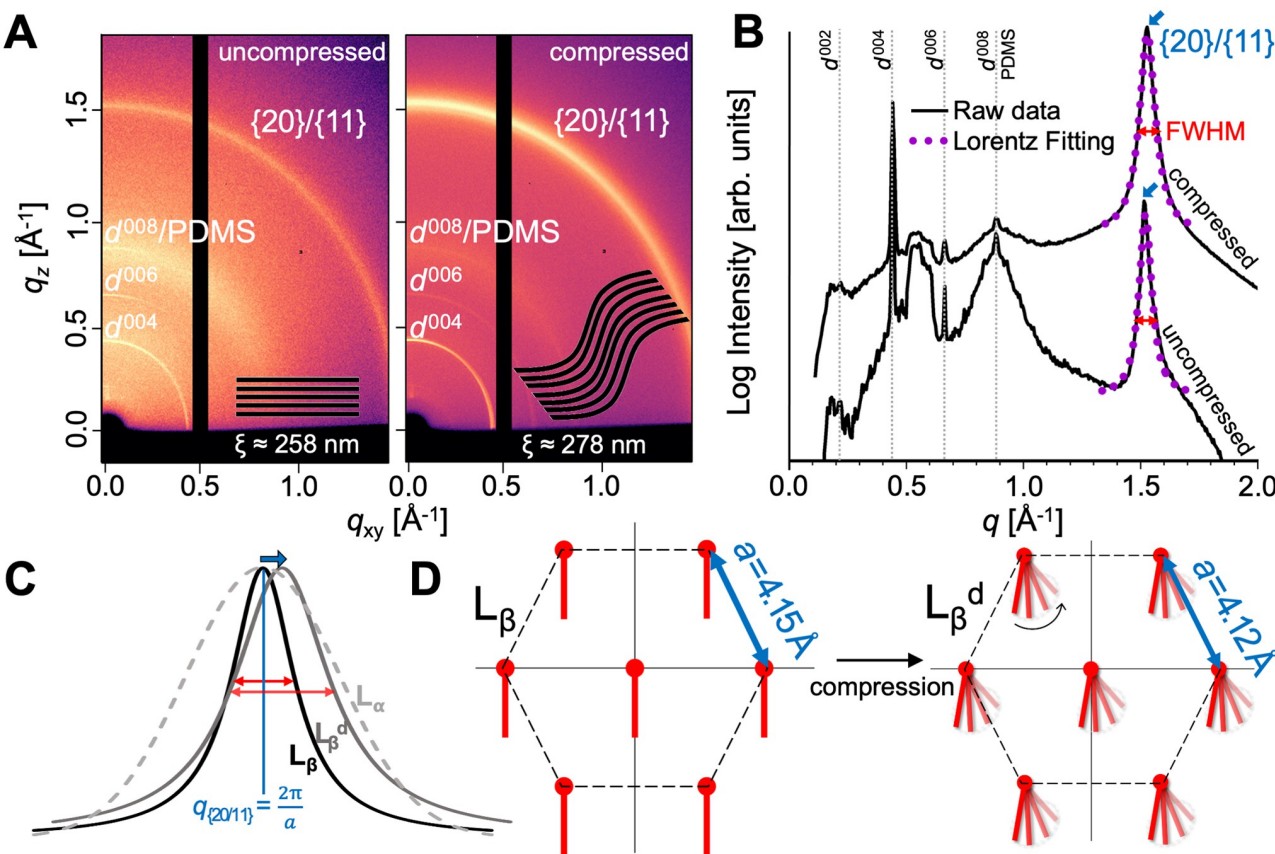

**Fig 2. X-ray scattering analysis of lipid films.** Reconfiguration of SLM films as a function of compression. A) 2D GIWAXS diffraction patterns of DPPC films before and after compression by 10%. The brackets denote a family of planes *hkl*. The inset are schematics of the changes in layer alignment corresponding with changes in correlation length $\xi$ induced in the DPPC SLM by compression. B) Linear integration of the GIWAXS data. C) Schematic of GIWAXS peak shift and widening related to the effect of compression at the hydrocarbon chain length scale. D) Schematic of DPPC hexagonal phase from GIWAXS data which displays changes in lattice spacing $a$ and increased disorder of lipid tails.

the GISAXS peaks yielding $q$ position and full width at half max (FWHM). Table A in S2 File summarizes the extracted structural parameters from the GISAXS and GIWAXS on both DPPC and DOPC under compressions of 0% (uncompressed), 5%, and 10%. The peak position in GISAXS is inversely related to bilayer-bilayer spacing $d$, while the peak position in GIWAXS is related to the molecular tail spacing $a$. The FWHM in GISAXS is inversely related to the correlation length for lipid bilayer alignment $\xi$, while the FWHM in GIWAXS is inversely related to the correlation length for molecular hydrocarbon chain crystallinity $\xi^\varphi$. Sharper peaks indicate higher alignment and crystallinity, respectively.

First, we will discuss the structure of DPPC before compression. The GIWAXS data in Fig 2A and 2B shows a sharp peak at $q \approx 1.5 \text{Å}^{-1}$ characteristic of lipids organized in the *gel* $L_\beta$ hexagonal phase, with lattice spacing $a = 4.15 \text{Å}$, and correlation length $\xi^\phi \approx 103 \text{Å}$. As a note, the peaks in Fig 2A and 2B) below $q \approx 1 \text{Å}^{-1}$ correspond to peaks arising from the lamellar phase in the GISAXS region. The peaks at $q \approx 1 \text{Å}^{-1}$ are combined with a broader peak arising from scattering of the PDMS substrate (Fig B in S2 File). The GISAXS peaks (Fig A and Table A in S2 File) reveal that the DPPC SLM films exhibited a lamellar bilayer stack structure, with an average inter-bilayer distance $d \approx 5.6$ nm. The anisotropic rings show that the bilayers are not completely aligned parallel to the substrate.

Fig 2C and 2D graphically summarizes the effect of compression on the hydrocarbon chain ordering. Under a 10% compression of the DPPC SLM, the GIWAXS and GISAXS diffraction peaks experience only small shifts, and significantly change their width (Fig 2C), indicating a reconfiguration of ordering, but not a change in phase. Specifically, the bilayer-bilayer spacing $d$ essentially does not change, while the lipid lattice spacing shifts from $a = 4.15$ Å before compression to $a = 4.12$ Å at 10% compression. This change in lattice parameter of <0.7% is much less than the strain being induced on the substrate, indicating that strain is primarily being relieved through other mechanisms, as we will discuss in the next section. Meanwhile, the order parameters both change significantly, $\xi$ from approximately 258 to 278 nm and $\xi^\varphi$ decreases from 103 to 83 Å. We interpret the increase in $\xi$ as an increase of domain size of about 20 nm i.e., approximate 5 more lipid bilayers aligned at the same orientation. The decrease in $\xi^\varphi$ is consistent with more disordered hexagonally packed lipids $L_\beta^d$ (Fig 2D), where the lipid chains span more orientations but do not transform to a fully disordered $L_\alpha$ liquid-like state.

To validate this conclusion, we also performed identical compression and analysis on DOPC, a lipid that naturally exists in the fully-disordered *fluid* $L_\alpha$ phase. Fig C in S2 File shows GIWAXS data for DOPC before and after 10% compression, Table A in S2 File contains the extracted lattice parameter, FWHM and domain size. We analyzed the data and compared the two lipids. Unlike DPPC, DOPC exhibits broad peaks consistent with a disordered liquid-like state. However, similarly to DPPC, these peaks broaden further upon compression, indicating an increase in disorder. In general, in both $L_\beta$ and $L_\alpha$ multi-layered PC membranes short-range $\xi^\varphi$ decreased in the compressed states, hence at the acyl chain interaction, order decreased upon compression.

The GISAXS and GIWAXS results show that mild mechanical load (10% compression) significantly affects lipid chain packing. In fact, the obtained effect is comparable to that of adding large amounts of cholesterol to $L_\beta$ DPPC membranes. Cholesterol affects the structure [39] phase behavior [40, 41], permeability [42, 43], mechanical properties [44, 45], molecular order [46, 47], orientation and intermolecular spacing [48, 49] of lipid membranes. Cholesterol causes positional/translational disordering of *gel* $L_\beta$ phase lipids while still maintaining a high degree of orientational ordering [50–54]. Here, we report a cholesterol-like effect on PC membranes induced solely by mechanical stress.

## Compression induced buckle delamination in lipid films

In the GISWAX data, the change in strain within the material is much less than the applied compression, indicating the substrate strain is being relaxed via other mechanisms. The most common mechanism of strain relief is Ruga, the process by which any elastic film under compression on a soft substrate adopts complex three-dimensional morphologies, such as wrinkles, creases, folds, and delamination buckles. These morphologies depend on the mechanical moduli of both the film and the substrate, along with the film thickness, adhesion of the film to the substrate and degree of compression, leading to a phase diagram where different morphologies exist in different parts of the parameter space [55]. Each morphology leads to different mechanics models which relate the geometrically measureable parameters to the material and interfacial properties.

We examined the influence of compression on the multiscale topography of DPPC SLM films and apply ruga mechanics models to extract the material moduli and adhesion (Fig 3). Fig 3A and 3B shows atomic force microscope (AFM) images of the topography of uncompressed and compressed lipid films at the microscale (Fig 3A) and nanoscale (Fig 3B). Before compression, the topography of the SLM film is mostly flat, with roughness on the scale of 1.9 ±0.7 nm, at both micro and sub-micro scales. However, as shown in schematic Fig 3C and AFM profile, compression induces the formation of multiscale wrinkling and buckle delamination increasing the roughness on the scale of 4.4 ± 0.9 nm with 14% compression. Both the wrinkles and buckle delamination run roughly perpendicular to the direction of the applied uniaxial strain.

The delamination buckles visible in the lipid film under compression are similar to what is commonly observed in high elastic modulus films with low adhesion to a substrate [56]. Thus,

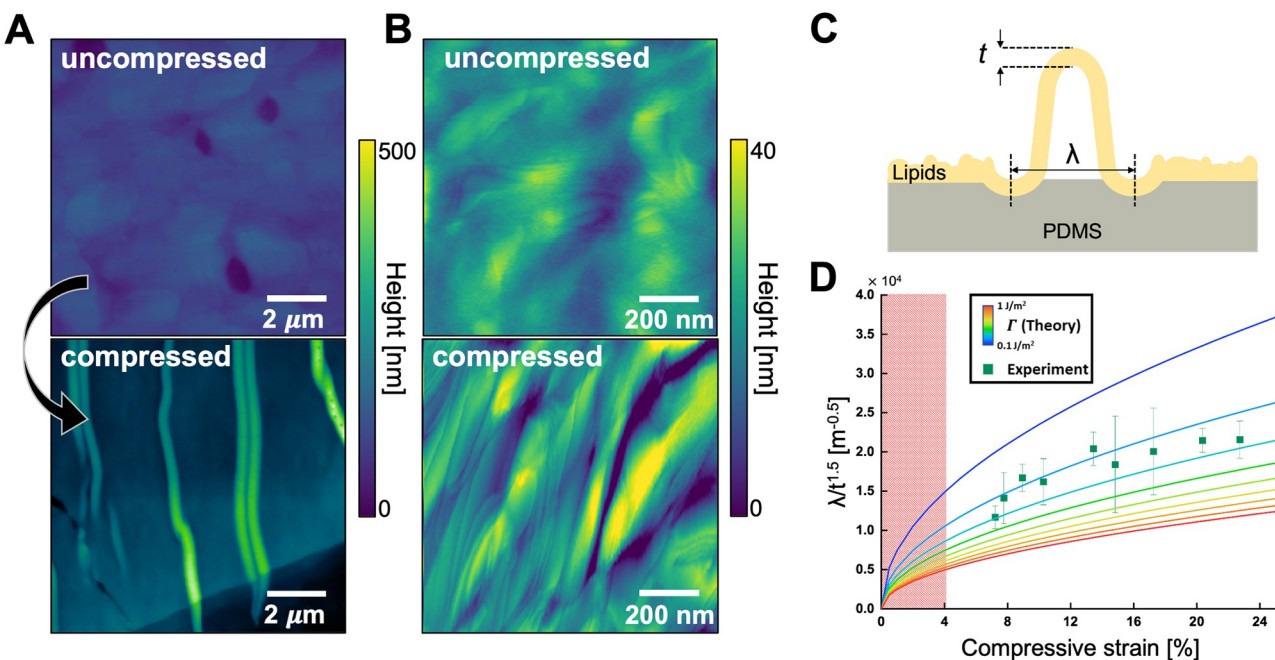

**Fig 3. Micro and nano scale deformation of DPPC SLM films after compression.** Micro (A) and sub-micro (B) scale topography maps of uncompressed and compressed lipid films. C) Schematic of wrinkling and delamination buckle of lipids film after compression where $t$ and $\lambda$ denote the thickness and width of the lipid film. D) Relationship between compression strain and $\lambda/t^{1.5}$. The colored lines on the graph correspond to the theoretical relationship based on continuum elastic model with different adhesion energies between DPPC SLM film and PDMS whereas the squares correspond to the experimental data measured in twelve different delamination buckles. The red shade indicates no observable delamination buckle onto lipid films after compression.

according to instability mechanics models, the geometry of the buckle (Fig 3C) is governed by the competition between bending, compression, and adhesion energy of the SLM on the substrate [57]. The equation governing the buckle delamination width is

$$\frac{\lambda}{t^{1.5}} = \sqrt{\frac{\pi^2 \bar{E}_f \epsilon}{\Gamma}}$$ (1)

where the material parameters such the adhesion energy $\Gamma$ and in-plane Young's modulus of the lipid film $\bar{E}_f$ are related to observable parameters like the substrate compressive strain $\epsilon$, delamination buckle width $\lambda$ and SLM film thickness $t$. This model assumes the strain in the thin film is negligible compared with the applied substrate strain, an assertion supported by the GIWAXS data. We describe the detailed derivation process in S3 File. By applying this equation, we extract the adhesion energy from the normalized buckle delamination shape.

Fig 3D plots the measured $\frac{\lambda}{t^{1.5}}$ under different compressive strain as well as the trends predicted using different adhesion values (colored lines). At each compression, we used AFM to measure the film thickness and width of buckles (Fig in S3 File). We assume a Young's modulus for DPPC of 42±16 MPa [58]. We limited measurements of the SLMs to the specific thickness range of 70–150 nm, because different film thicknesses of DPPC show variations in surface roughness which affect accurate comparison [59]. We compare this theoretical calculation (colored lines) against the experimental data (squares). By fitting the experimental data, the model reveals that the adhesion energy between the DPPC multilayer film and PDMS is $0.22 \pm 0.08$ J/m$^2$. For comparison, the adhesion energy between two DPPC bilayers in water is 0.15 mJ/m$^2$ [60] which is three orders of magnitude smaller than that of DPPC multilayered film on PDMS. This discrepancy may be explained by different environment (water vs air) and different film thickness (4.6 nm for DPPC bilayer [60] vs 70–150 nm for DPPC SLM film). A better comparison is to other materials dominated by dry van der Waals adhesion. For example, the adhesion energy for atomically thin graphene and SiO$_2$ is 0.096–0.45 J/m$^2$ [61–64], on the same order with what we measure for the DPPC SLM films.

Comparing these results back to the original Ruga mechanics phase diagrams [56], the values for adhesion and Young's modulus and compression would predict that the SLM films are in a regime of buckle delamination and wrinkles, as observed. These results suggest the DPPC SLM film behaves as an elastic film right after compression. However, over the time scale of 24 hours after compression, we observed a time dependent relaxation of the surface patterns of the DPPC SLM films (Figs A and B in S4 File) This relaxation in morphology suggests that there is a inelastic restructuring of the film stress over time, which is consistent with the increased molecular disorder observed in the GIWAXS data. To account for these relaxations, all the morphology and optical measurements we presented in the main text were acquired within an hour after inducing compression.

## Spatial distribution of hydrocarbon lipid chain packing upon compression

We used confocal microscopy and fluorophores specified to explore how molecular-scale lipid reconfiguration is spatially distributed and how it relates to the microscale morphologies under compression (Fig 4). To probe the spatial heterogeneity of the lipid ordering, we incorporate Laurdan flourophores into the SLM. Laurdan is an environmentally sensitive fluorescent probe commonly used to image lipid systems because changes in the environmental polarity, such as those induced by changes in lipid tail order, result in measurable shifts in the spectral emission. Specifically, higher order leads to a blueshift and lower order leads to a

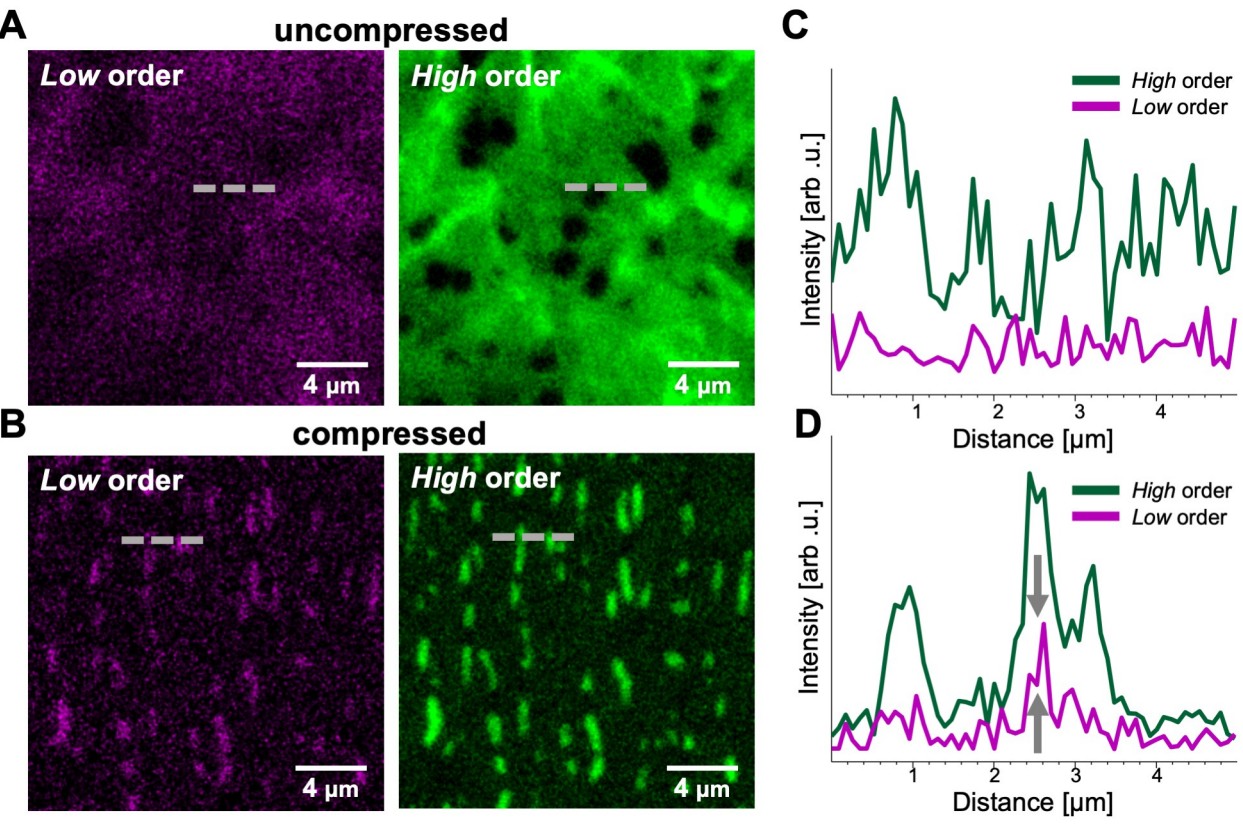

**Fig 4. Optical analysis of hydrocarbon chain order of DPPC SLM films upon compression.** A) Confocal laser scanning microscopy (CLSM) images of DPPC-Laurdan SLM films adsorbed onto PDMS before and B) after compression by 20%.Low–ordered hydrocarbon chains fluorescence in magenta (excitation laser at 405 nm/detection wavelength 475–700 nm) and high–ordered hydrocarbon chains fluorescence in green (excitation laser at 405 nm/detection wavelength 400–-456 nm). C) and D) 1D intensity profiles along the dash gray line in (A) uncompressed and (B) compressed films respectively of each Laurdan dye channel (high–ordered hydrocarbon chains fluorescence in green and low–ordered hydrocarbon chains fluorescence in magenta).

redshift. To the best of our knowledge, using Laurdan optical probe to monitor compression–induced changes in lipid order of multilayered films has not been reported.

Fig 4A shows two–dimensional cross–sectional confocal laser scanning microscopy (CLSM) images of DPPC SLM onto PDMS labeled with 0.3 mol% Laurdan. The top and bottom sets of images are from the same sample before (uncompressed) and within 15 minutes after 20% compression. The magenta images are the red-shifted emission from 475–700 nm, corresponding with lower order in the host SLM. The green images are the blue-shifted emission from 400–456 nm, corresponding with higher order in the host SLM. Fig 4B plots the emission intensity along the dashed lines in each of the images, comparing the relative intensity for the *high* and *low* order in the uncompressed (top) and compressed states (bottom), respectively. Because of the difficulty of identifying the exact region of interest across images, the before and after sections show different regions, representative of the overall behavior. Fig A in S5 File shows additional images and intensity profiles from other regions.

In the uncompressed SLM, shown in the top images of Fig 4A and 4B, both the *high* order green channel and *low* order magenta channel show distinct high-intensity domains. The domains in both channels mostly overlap, suggesting that the changes in signal intensity come from variations in thickness or Laurdan concentration rather than significant variations in local lipid-chain order. The overall morphology of supported lipid domains and the higher

intensity from the green channel are consistent with that observed for the solid-like, ordered *gel* L$_\beta$ phase in DPPC films. [65–67] We also calculated the effect of thin-film interferometry on the signal intensity, and show that it does not significantly affect the results (S5 File).

As films undergo compression, a clear modification in film morphology occurs. The developed buckle delamination patterns are analogous to those observed under AFM (Fig 3A and 3B). Interestingly, higher intensity green fluorescence is co-localized at the wrinkled regions. This is consistent with the idea that lipid chains with more order preferentially distribute onto the buckle delaminated regions. The intensity profile of both channels is plotted in uncompressed vs. compressed films in Fig 4B. We observe that even though the green channel signal dominates, as films are compressed there is a noticeable increase of magenta channel (grey arrows) signal arising from less well-packed chains. It is notable that, the intensity signals from *low* and *high* order phases could be affected by interference between the two detection channels. Fig B in S5 File shows the Laurdan dye emission spectra exhibit some level of overlap between the peaks for *high* and *low* order systems. Hence, it is possible that there is crosstalk between the emission signal of each phase. Nevertheless, the CLSM results align qualitatively with what was observable by WAXS and molecular dynamics simulations. Taken together, three independent experiments serve as confirmation that compressing SLMs leads to an increase of disorder of some alkyl chains while others remain in the ordered state and preferentially distribute at the top of the buckle-delaminated wrinkle.

## Compression decreases chain order parameters by inducing nanoscale undulations

We used molecular dynamics to simulate the effect of compressive strain on multilayered lipid membranes (Fig 5). In living systems, multilayered lipid structures are often supported onto polymeric flexible substrates such as actin filaments and the extracellular matrix, indicating the adaptive yet robust nature of these systems. Shown in Fig 5A, we simulated the structure of *gel* L$_\beta$ phase DPPC multilayered membranes both in the uncompressed state and under compressive strains of 5% and 10%. The color coding in the images indicates the lipid tilt angle (orientation of the molecule) with respect to the membrane normal (z-axis). Lipid directors have been previously used [68] to represent the direction of individual lipid molecules. We implemented a similar approach in this study (Fig A in S6 File) to compute the lipid tilts. The simulations show a formation of nanoscale undulations in the multilayered lipid membranes when subjected to compressive strains. Further, Fig 5B shows the corresponding histograms of lipid tilt angles for a DPPC multilayered membrane under compression. The histograms show that with increase in compressive strain the undulations increase, resulting in higher lipid tilts. This is indicated by a shift in the peak values from 10˚(uncompressed) to 13.5˚(5% strain) and 22.5˚(%10 strain). (Fig B in S6 File) shows the same data for a L$_\alpha$ phase multilayered DOPC membrane and similar trends were observed. We note that the simulation is periodic along the z-axis, so the patterns repeat in each bilayer. The membrane-substrate interactions have not been accounted for in the simulations as the focus was to probe the effects of compression alone. Previous studies have explored the substrate effects on the morphological and dynamical properties of supported lipid membranes [69–71]. For supported lipid bilayers, a change in the lipid properties has been reported only in the inner (closest to the substrate) lipid leaflet, whereas the outer leaflet properties were similar to a free-standing bilayer [71]. Hence, for the multilayered membranes considered in this study, the substrate effects are insignificant, and only the lipid-lipid interactions dictate the membrane properties.

We elucidate the effect of compression on lipid order by computing the order parameter in the simulations for both DOPC and DPPC under each compression, and listed the average

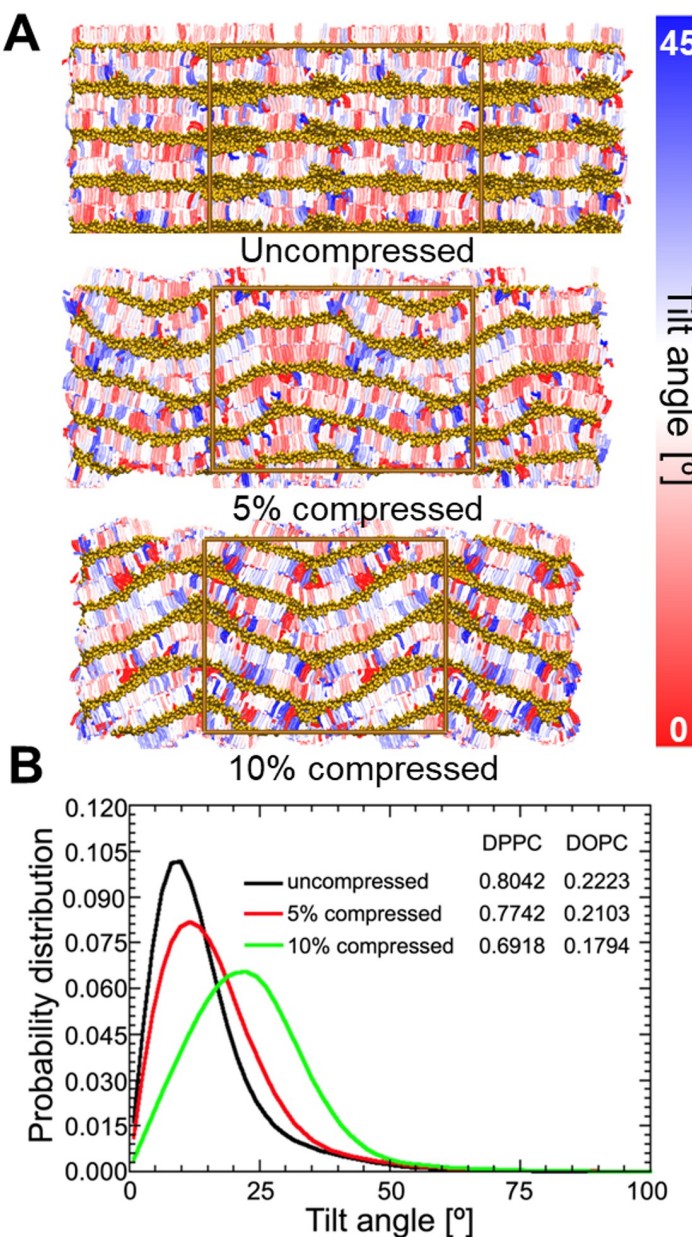

**Fig 5. Effect of compression on lipid tilt.** A) Color coded representation of the simulated DPPC systems at three compressive strains. The color represents lipid orientation (tilt) angle with respect to z-axis. The box represents the simulation unit cell. B) The tilt angle distributions of the $L_\beta$ phase DPPC lipids. The inset shows the average order parameter values.

tail-order parameter values (inset of Fig 5B). The second-rank order parameter is defined as $P_2 = \frac{1}{2}(3\cos^{2\theta-1})$, where $\theta$ is the angle made by the bonds between the coarse-grained beads with respect to the membrane normal. Fig B in S6 File shows how we computed these average order parameter values. The order parameter varies between $P_2 = 1$ (perfect order along the normal direction) and $P_2 = -0.5$ (perfect order perpendicular to the normal), with $P_2 = 0$ indicating random orientations. Similar to the experimental results, both DPPC and DOPC, demonstrated an increase in lipid tail disorder upon the application of compressive strain.

The order parameter computed here depends on a) the orientation (tilt) of individual lipid molecules with respect to the membrane normal, and b) within each lipid, the angle made by the lipid bonds with respect to the orientation of the respective lipid molecule, and both factors can contribute to the observed reduction in order-parameter with strain. In order to isolate and understand the individual contributions of these factors in the observed reduction in lipid order, we renormalized the order-parameter with respect to the lipid directors (instead of z-axis). This approach removes the effect of the changing lipid orientations (due to undulations). Interestingly, these order parameter values (Fig B in S6 File) did not show any significant differences between the uncompressed and compressed cases. The renormalization shows that the observed reduction in lipid order is a direct result of the formation of nanoscale undulations which cause the lipids to attain tilts with respect to the membrane normal. There is negligible change in lipid tail conformational order with strain. This result is completely consistent with the GIWAXS data (Fig 2D) where compression leads to broadening of the diffraction peak characteristic for lipid alkyl chain packing.

Next, to understand the effect of strain on the correlation in these lipid tilts across layers, we computed the inter-layer correlations in lipid tilt (angle between the lipid director and z-axis). The correlation was computed as $\frac{<\theta(l1).\theta(l2)>}{<\theta(l1)\theta(l1)>}$, where $\theta$ is the instantaneous angle of lipid orientation with respect to the z-axis. The layers are laterally divided into segments (50x50) and correlations between layers $l1$ and $l2$ are computed separately for pairs of lipids in each segment. Here, $<>$ denotes an average over the different pairs, segments and time frames. Similar to experimental results for both DPPC and DOPC, the inter-layer correlation in lipid orientations increase with compressive strain (Fig C in S6 File). However, since the simulated systems contain only four independent layers (a choice made to reduce computational cost) the correlations could be computed for a maximum of two layers across.

## Conclusion

Lipid membranes are highly dynamic and mechanically active biointerfaces capable of responding and adapting to stress and stimuli. This adaptability is manifested in natural systems that have oscillatory mechanical loads such as the example of inflating/deflating alveoli. In this article we have investigated the structural and mechanical evolution of multilayered films composed of the major lipid in pulmonary membranes both in *gel* $L_\beta$ (DPPC) and *fluid* $L_\alpha$ (DOPC) phases as a result of subjecting them to compressive stress. The findings suggest that compression reconfigures the lipid films at multiple length scales impacting their elastic behavior. First, compressive stress induced the formation of morphological features such as delamination buckling leading to larger domain size of the lipid bilayers within the multilayered film, without altering the inter-bilayer distance. Second, compression reduced the liquid crystallinity of the lipid hydrocarbon tails increasing the domain size of conformational changes of C-C bonds i.e., higher disorder and mobility, while slightly decreasing the separation between tails. This was a direct result of the formation of nanoscale undulations across the lipid multiple-layers. We calculated the adhesion energy between a $L_\beta$ multilayered film and PDMS to be 0.22±0.08 J/m$^2$ which resulted akin to the energies between atomically thin graphene and SiO$_2$. Then, we reported that $L_\beta$ multilayers behave as elastic films upon compression. However, we observe a time-dependent relaxation of the surface features suggesting that the films exhibit inelastic signatures over time. We propose that this cooperative short- and long-range reconfiguration of lipid multilayered films under compression results as a mechanism to comply with stress and substrate topography. Consequently, a deeper understanding of lipid multiple-layer structure and mechanics under stress will enable tailoring lipid film

behavior to achieve functionality; this constitutes a strategy for developing mechanically reconfigurable multifunctional surfaces for diverse applications such as and bioelectronics.

## Materials and methods

### Materials

1,2-dipalmitoyl-sn-glycero-3-phosphocholine (DPPC), 1,2-dioleoyl-sn-glycero-3-phospho-choline (DOPC), fluorescent probe 16:0 Liss Rhod-PE and Laurdan (6-dodecanoyl-2-dimethyl-laminonaphthalene) were purchased from Avanti Polar Lipids (Alabama, USA) suspended in chloroform. Polydimethylsiloxane (PDMS) was purchased from Gel-Pak (part No. PF-60-X4) (California, USA). The Young's modulus of PDMS was measured by using a custom-made stretching platform.

### X-ray analysis

Lipid films were adsorbed onto the pre-stretched PDMS substrates ($1.2 \times 1.2$ $cm^2$). Approximately 100 $\mu$L of the stock solution of lipids (20 mM) in ethanol was drop-casted onto the substrate and allowed to dry in air. Later, the samples were dried-vacuum for at least two hours and the samples were incubated at 50 ˚C in high relative humidity (RH $\approx$ 95%) for 4 hours. Finally, the pre-stretched samples were released before characterization. To characterize the lipid film structure and orientation, GISAXS/GIWAXS was conducted in a custom built (with Forvis Technologies, Santa Barbara) equipment composed of a Xenocs GeniX3D Cu $\kappa\alpha$ ultra-low divergence X-ray source (1.54 Å/8 keV), with a divergence of 1.3 mrad. Multiple measurements were carried on the same sample in the uncompressed and compressed states, varying the incidence angle to determine the most appropriate operating angle. The sample-to-detector distance was calibrated using a silver behenate powder standard. The 2D diffraction data were radially averaged upon acquisition on a Pilatus 300 K 20 Hz hybrid pixel detector (Dectris). The 2D diffraction patterns were reduced to 1D plots using SAS2D [72].

### Atomic force microscopy

For the preparation of lipid multilayered membranes, stock solutions of lipids in a mixture of methanol and chloroform 1:1 were mixed to a final concentration of 20 mM. A volume of 70 $\mu$L of the lipid organic solutions was pipetted onto pre-stretched PDMS substrates ($1.2 \times 1.2$ $cm^2$) which afterwards was immediately accelerated to 3000 rpm for 30 s using a spin coater (VTC-100A, MTI Corporation, California, USA). Afterwards, the pre-stretched samples were vacuum-dried for at least 2 hours. Subsequently, the samples were mounted and scanned. Atomic force microscopy experiments were performed in a Tosca 400 (Anton Paar, Ballerup, Denmark) and a Cypher AFM (Asylum Research-Oxford Instruments, California, USA). Topography imaging was performed in intermittent contact (tapping mode) in air using Si3N4 cantilevers (AC240TS-R3, Oxford Instrumets, California, USA) with nominal spring constant of 2 N/m.

### Confocal microscopy

To prepare the dry spin–coated lipid multilayered films on pre–streched PDMS substrate, we used a stock solution of 20mM DPPC containing 0.3mol% Laurdan in methanol/chloroform (1:1 volume ratio). A volume of 70 $\mu$L of this lipid stock solution was pipetted onto pre–stretched PDMS substrates ($1.2 \times 1.2$ $cm^2$) and immediately thereafter spun on a VTC–100A (MTI Corporation, California, USA) spin–coater at 3000 rpm for 30 s. The pre–streched sample was then placed under vacuum for at least 2 hours to ensure complete removal of the

organic solvents. Finally, the pre–streched and released samples were then placed on the LSM 800 confocal microscope (Carl Zeiss Microimaging, Jena, Germany). Imaging was performed in open air, room RH and temperature using a Plan–Apochromat 40 × /0.9 Pol M27 objective lens. With Laurdan probe, a single excitation wavelength λex = 405 nm was used in line scanning mode to allow simultaneous collection of high–ordered hydrocarbon tails (green color) and low–ordered hydrocarbon tails (magenta color) images. The emitted light was split using gradiated beam splitters to 2 detectors: 400 nm to 456 nm (high–ordered hydrocarbon tails), and 475 nm to 700 nm (low–ordered hydrocarbon tails). The same gain, laser intensity, and the exposure time was applied to each channel.

## Computer simulations

**Forcefield.** We used coarse-grained molecular dynamics simulations to study the effect of strain on the properties of multi-lamellar phases of DPPC and DOPC lipids. The experimental systems consist of multi-lamellar stacks of lipids with a thin layer of water in between. The lipids have been modelled using the MARTINI coarse-grained force-field, which has been previously used to successfully model membrane processes [73, 74]. Specifically, the implicit solvent version of MARTINI called dry-MARTINI—where the interaction between the beads are adjusted to mimic the presence of water—was used to model the lipids and the interlayer water. Dry-MARTINI has been previously shown to be as accurate as the explicit solvent versions at reproducing the experimental membrane properties, while being computationally less expensive [75].

## Simulation systems and protocols

Two kinds of simulations were performed—uncompressed simulations in the NPT ensemble, and compressed simulations in the $NP_zAT$ ensemble. In all the simulations, a velocity rescaling thermostat [76] was used to maintain the temperature, and a Parrinello-Rahman barostat [77] was used for pressure coupling. Initially, DPPC and DOPC bilayers (1000 lipids per monolayer) were equilibrated under the NPT ensemble (T = 300K and P = 1atm). Four such pre-equilibrated bilayers were stacked on top of each other with periodic boundary conditions in all three directions, to generate a lamellar phase. Four layers were used to facilitate studying the interlayer correlations discussed in the results section.

The resultant system was simulated for 1 $\mu$s under the NPT ensemble to equilibrate the lamellar phase. The equilibrated system was compressed along the x-direction at a steady rate (0.00184 $A^0/ps^{-1}$ and 0.001675$A^0/ps^{-1}$ for DOPC and DPPC, respectively) by 5% and 10% of the initial box length to study the effect of strain on the lamellar phase. Post compression, each of the strained systems were re-equilibrated for 1 $\mu$s under the $NP_zAT$ ensemble (constant pressure along z and constant x-y area at 300K). The equilibrated un-strained and strained systems were simulated for additional 200ns to collect data for analysis. All the simulations were performed in the GROMACS [78] simulation package.

## Supporting information

**S1 File. Platform for inducing compression of lipid films.**
(PDF)

**S2 File. X-ray scattering data analysis.**
(PDF)

**S3 File. Derivation of adhesion energy between supported lipid films and PDMS.**
(PDF)

**S4 File. Time-dependant relaxation of supported lipid films.**
(PDF)

**S5 File. Laurdan data analysis.**
(PDF)

**S6 File. Simulations.**
(PDF)

## Acknowledgments

We acknowledge the use of parallel computing resource Lonestar6 provided by the Texas Advanced Computing Center (TACC) at The University of Texas at Austin. The authors thank Dylan Steer for providing the Laurdan spectral analysis for pure DPPC and DOPC lipids.

## Author Contributions

**Conceptualization:** Marilyn Porras-Gómez, Mohan Teja Dronadula, Arend van der Zande, Cecília Leal.

**Funding acquisition:** Arend van der Zande, Cecília Leal.

**Investigation:** Marilyn Porras-Gómez, Hyunchul Kim, Cecília Leal.

**Methodology:** Marilyn Porras-Gómez, Hyunchul Kim, Mohan Teja Dronadula, Nurila Kambar, Christopher J. B. Metellus.

**Resources:** Narayana R. Aluru, Arend van der Zande, Cecília Leal.

**Software:** Mohan Teja Dronadula, Narayana R. Aluru.

**Supervision:** Arend van der Zande, Cecília Leal.

**Writing – original draft:** Marilyn Porras-Gómez, Hyunchul Kim, Mohan Teja Dronadula, Nurila Kambar.

**Writing – review & editing:** Marilyn Porras-Gómez, Narayana R. Aluru, Arend van der Zande, Cecília Leal.

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
