## [Decision Letter · Decision Letter 0]

7 Oct 2022

PONE-D-22-25179Multiscale compression-induced restructuring of stacked lipid bilayers: from buckling delamination to molecular packingPLOS ONE

Dear Dr. Leal,

Thank you for submitting your manuscript to PLOS ONE. After careful consideration, we feel that it has merit but does not fully meet PLOS ONE’s publication criteria as it currently stands. Therefore, we invite you to submit a revised version of the manuscript that addresses the points raised during the review process. Specifically although the manuscript was well received, concerns over the laurdan experiments, and how membrane-substrate interactions were accounted for in the simulations you conducted need to be addressed before the manuscript can be accepted for publication.

We look forward to receiving your revised manuscript.

Kind regards,

Colin Johnson, Ph.D.

Academic Editor

PLOS ONE

Journal Requirements:

2. Please note that PLOS ONE has specific guidelines on code sharing for submissions in which author-generated code underpins the findings in the manuscript. In these cases, all author-generated code must be made available without restrictions upon publication of the work. Please review our guidelines at https://journals.plos.org/plosone/s/materials-and-software-sharing#loc-sharing-code and ensure that your code is shared in a way that follows best practice and facilitates reproducibility and reuse. New software must comply with the Open Source Definition.

"This research was primarily supported by the National Science Foundation through the University of Illinois Urbana-Champaign Materials Research Science and Engineering Center DMR-1720633, and partially under National Science Foundation grant number 2140225, and the Office of Naval Research grant number N00014-21-1-2029. We acknowledge the use of parallel computing resource Lonestar6 provided by the Texas Advanced Computing Center (TACC) at The University of Texas at Austin. "

"AZ, NA

NSF DMR-1720633

National Science Foundation through the University of Illinois Urbana-Champaign Materials Research Science and Engineering Center

https://mrsec.illinois.edu

AZ, NY

NSF 2140225

National Science Foundation

https://www.nsf.gov/

CL

ONR N00014-21-1-2029

Office of Naval Research

https://www.nre.navy.mil/

Reviewers' comments:

Reviewer's Responses to Questions

**Comments to the Author**

1. Is the manuscript technically sound, and do the data support the conclusions?

Reviewer #1: Yes

2. Has the statistical analysis been performed appropriately and rigorously? 

Reviewer #1: Yes

3. Have the authors made all data underlying the findings in their manuscript fully available?

Reviewer #1: Yes

4. Is the manuscript presented in an intelligible fashion and written in standard English?

Reviewer #1: Yes

5. Review Comments to the Author

Reviewer #1: This is an interesting and timely piece of work. As the authors state. there aren't many studies on mechanics of multilayer membranes. Below are some minor comments:

1) Can the authors state the controlled release increments of the subtrate strain on page 3

2) explain the color coding in Fig. 1

3) typos on line 139 and 155

4) revise line 198- is that the adhesion energey between the individual lipid membranes?

5) Laurdan experiments: I am not sure I see the high-intensity distinct domians on Fig. 4A,C (line 242); There seems to be crosstalk between the two channels as they both show fluorescence at the same regions in the membrane. For this reason I am not convinced that the increase in the magenta channel on 4B is due to increasing disorder. Can the authirs use spectral imaging instead?

6)The conclusion of the paper is that nanoscale mebrane undulations resulting from the compression are the main trigger for the change in membrane order. Since the delamination effects and buckling are dependent mainly on the membrane-susbtarte intercation, how is this intercation taken into account in the simulations on fig. 6? The authors may include a comment on the biological relevance of this intercation?

6. PLOS authors have the option to publish the peer review history of their article (what does this mean?). If published, this will include your full peer review and any attached files.

Reviewer #1: No

---

## [Author Response · Author response to Decision Letter 0]

1 Nov 2022

We are thankful to the Editor for giving us the opportunity to revise our manuscript. The Reviewer recommended minor revisions and provided important points for improvement. We clarify previously confusing or incomplete statements and expanded on our results and discussion to provide a revised document that fully addresses all concerns/comments/and suggestions raised by the Reviewer. In the report below we answer (in boxed text) to these concerns (in bold text) point-by-point and provide the corresponding changes made to the manuscript and the supporting material. We highlighted the changes made to the original version in the file labeled 'Revised Manuscript with Track Changes'.

Reviewer #1: This is an interesting and timely piece of work. As the author’s state. There aren't many studies on mechanics of multilayer membranes. Below are some minor comments:

We thank the Reviewer for acknowledging the importance and potential impact of our work. We are particularly thankful for their careful assessment and constructive criticism that guided us to produce an improved version of our initial manuscript. 

1) Can the authors state the controlled release increments of the substrate strain on page 3.

We thank the Reviewer for identifying a lack of explanation here. We have expanded the line to quantify the percentages we used to pre-strain the PDMS substrates. Lines 83-85 in page 3 now read: “Finally, we released the substrate pre-strain by controlled increments of 5, 10 or 20% to induce compression on the lipid film.”

2) Explain the color coding in Fig. 1.

We appreciate the Reviewer’s attention to detail. We have added an additional component to Fig 1 to enlarge a section of the simulated lipid multilayers with the objective of providing clarification for the color coding. We have also expanded the caption of Fig 1.

Fig 1. Substrate-mediated reconfiguration of lipid multilayered films. Schematics and simulations (discussed further in Fig 5) of the multiscale structure of multilayered lipid films on a deformable substrate before (A) and after (B) compression. The schematic in the right expands on a section of the lipid multilayered system to highlight a bilayer where headgroups are represented in red and tails in yellow.

3) Typos on line 139 and 155

We thank the Reviewer for indicating the typos. Both have been corrected in the corresponding lines. 

4) Revise line 198- is that the adhesion energy between the individual lipid membranes?

We appreciate the Reviewer for requesting clarification. The Reviewer is correct, the adhesion energy cited in line 198 correspond to adhesion between two lipid bilayers, and we have edited the sentence to clarify it, which now reads: “For comparison, the adhesion energy between two DPPC bilayers in water is 0.15 mJ/m2.”

5) Laurdan experiments: I am not sure I see the high-intensity distinct domains on Fig. 4 A, C (line 242); There seems to be crosstalk between the two channels as they both show fluorescence at the same regions in the membrane. For this reason, I am not convinced that the increase in the magenta channel on 4B is due to increasing disorder. Can the authors use spectral imaging instead?

We are thankful that the Reviewer brings up this point. The crosstalk between the two detection channels was addressed by adding Laurdan spectroscopy data (new Figure B in S6 Supporting Information). There is a constrain in emission detection due to narrow spectral separation between low and high order domains as suggested by the Reviewer. This caveat has been addressed on the main manuscript in page 7, lines 258-264: "It is notable that, the intensity signals from low and high order phases could be affected by interference between the two detection channels. Figure B in S5 Supporting Information shows the Laurdan dye emission spectra showing some level of overlap between the peaks for high and low order systems. Hence, it is possible that there is crosstalk between the emission signal of each phase. Nevertheless, the CLSM results align qualitatively with what was observable by GIWAXS and molecular dynamics simulations.”

Here we provide more details of the new Figure B in S5 Supporting Information:

We performed Laurdan spectral analysis in ambient environment for pure DPPC and DOPC lipids, which are representative of high and low order membranes at room temperature, respectively. Samples were excited with Xenon lamp excitation at λex = 350 nm (bandpass: 20 nm) and the emission spectra was collected in n=3 measurements. DPPC at low temperature (25 °C) is assembled in a bilayer where hydrocarbon chains have a high degree of order yielding a Laurdan emission spectra with a maximum at ca. 443 nm. DOPC at 25 °C (above its transition temperature) comprises hydrocarbon tails with more conformational freedom yielding a peak with a maximum at ca 480 nm.

Figure B in S5 Supporting Information. Emission spectra of 1 mol% Laurdan in 50 mM DPPC and DOPC samples at 25 °C. Both samples were excited at 350 nm (20 nm bandpass) with Xenon lamp and corresponding spectra was collected using Synergy Neo 2 microplate reader (Biotek). DPPC has a high order phase Laurdan emission spectra with a peak at ca. 443 nm (colored in green). DOPC has a low order phase with a peak at ca. 480 nm (colored in magenta).

6) The conclusion of the paper is that nanoscale membrane undulations resulting from the compression are the main trigger for the change in membrane order. Since the delamination effects and buckling are dependent mainly on the membrane-substrate interaction, how is this interaction taken into account in the simulations on fig. 6? The authors may include a comment on the biological relevance of this interaction.

We appreciate the Reviewer for asking clarification on this matter. In this study, molecular simulations have been performed to probe the effect of compression on supported multilayered membranes at the nanoscale. In living systems, multilayered lipid structures are often supported onto polymeric flexible substrates such as actin filaments and the extracellular matrix, indicating the adaptive yet robust nature of these systems. Previous studies have explored the nanoscale effects of substrate on membrane properties [new references 69-71]. For supported lipid bilayers, a change in lipid morphology and dynamics has been reported only in the inner (closest to the substrate) lipid leaflet, whereas the outer leaflet properties were similar to a free-standing bilayer [71]. Hence, for the multilayered membranes considered in this study, the substrate effects are insignificant, and only the lipid-lipid interactions dictate the membrane properties. 

Changes have been made in the main manuscript (page: 8, lines: 271-273, 288-296) to explain this.

---

## [Decision Letter · Decision Letter 1]

24 Nov 2022

Multiscale compression-induced restructuring of stacked lipid bilayers: from buckling delamination to molecular packing

PONE-D-22-25179R1

Dear Dr. Leal,

We’re pleased to inform you that your manuscript has been judged scientifically suitable for publication and will be formally accepted for publication once it meets all outstanding technical requirements.

Kind regards,

Colin Johnson, Ph.D.

Academic Editor

PLOS ONE

Additional Editor Comments (optional):

Reviewers' comments:

Reviewer's Responses to Questions

**Comments to the Author**

1. If the authors have adequately addressed your comments raised in a previous round of review and you feel that this manuscript is now acceptable for publication, you may indicate that here to bypass the “Comments to the Author” section, enter your conflict of interest statement in the “Confidential to Editor” section, and submit your "Accept" recommendation.

Reviewer #1: All comments have been addressed

2. Is the manuscript technically sound, and do the data support the conclusions?

Reviewer #1: Yes

3. Has the statistical analysis been performed appropriately and rigorously? 

Reviewer #1: Yes

4. Have the authors made all data underlying the findings in their manuscript fully available?

Reviewer #1: Yes

5. Is the manuscript presented in an intelligible fashion and written in standard English?

Reviewer #1: Yes

6. Review Comments to the Author

Reviewer #1: I am happy with how the authors have addressed my comments and I fully support the publication of the manuscript.

7. PLOS authors have the option to publish the peer review history of their article (what does this mean?). If published, this will include your full peer review and any attached files.

Reviewer #1: No

---

## [Editor Report · Acceptance letter]

1 Dec 2022

PONE-D-22-25179R1 

Multiscale compression-induced restructuring of stacked lipid bilayers: from buckling delamination to molecular packing 

Dear Dr. Leal:

I'm pleased to inform you that your manuscript has been deemed suitable for publication in PLOS ONE. Congratulations! Your manuscript is now with our production department. 

Kind regards, 

on behalf of

Dr. Colin Johnson 

Academic Editor

PLOS ONE